# Rift Valley Fever Virus—Infection, Pathogenesis and Host Immune Responses

**DOI:** 10.3390/pathogens12091174

**Published:** 2023-09-19

**Authors:** Niranjana Nair, Albert D. M. E. Osterhaus, Guus F. Rimmelzwaan, Chittappen Kandiyil Prajeeth

**Affiliations:** Research Center for Emerging Infections and Zoonoses, University of Veterinary Medicine Hannover, Foundation, 30559 Hannover, Germany; niranjana.nair@tiho-hannover.de (N.N.); albert.osterhaus@tiho-hannover.de (A.D.M.E.O.); guus.rimmelzwaan@tiho-hannover.de (G.F.R.)

**Keywords:** Rift Valley Fever Virus (RVFV), NSs gene, immunity, vaccine

## Abstract

Rift Valley Fever Virus is a mosquito-borne phlebovirus causing febrile or haemorrhagic illness in ruminants and humans. The virus can prevent the induction of the antiviral interferon response through its NSs proteins. Mutations in the NSs gene may allow the induction of innate proinflammatory immune responses and lead to attenuation of the virus. Upon infection, virus-specific antibodies and T cells are induced that may afford protection against subsequent infections. Thus, all arms of the adaptive immune system contribute to prevention of disease progression. These findings will aid the design of vaccines using the currently available platforms. Vaccine candidates have shown promise in safety and efficacy trials in susceptible animal species and these may contribute to the control of RVFV infections and prevention of disease progression in humans and ruminants.

## 1. Introduction

Bunyaviruses are among the largest orders of RNA viruses comprising nine virus families and are transmitted by arthropods or rodents. Viruses belonging to the Phlebovirus genus of the Phenuiviridae infect multiple hosts and have a major impact on agriculture, animal husbandry and human health. [1,2,3,4,5]. Phleboviruses contain more than 60 different species and are predominantly transmitted by arthropods such as ticks, mosquitoes and sandflies [6]. Rift Valley Fever Virus (RVFV) is one of the phleboviruses and was first identified in Rift Valley of Africa in the early 1930s, initially as a causative agent of neonatal mortality and abortion storms in sheep and, subsequently, also as a cause of febrile illness in man [7,8]. Primarily domesticated ruminants such as sheep, goat, cattle and ungulates like camels were considered to be most susceptible, however, there is also serological evidence of infection in several other wild animals such as impalas, giraffes, pigs, warthogs, etc. [9,10,11,12,13]. 

RVFV is transmitted among ruminants by mosquitoes such as *Aedes*, *Anopheles* and *Culex* species, with large outbreaks occurring frequently during heavy rainfall [14,15,16]. Mosquito-borne transmission among humans is rare and the major mode of transmission is by direct contact with infected blood, carcass or by the consumption of raw milk from infected ruminants [17,18,19]. Signs of infection in ruminants generally start with fever and could manifest into respiratory illness, hepatic necrosis, abortions, stillbirths and death [20]. In humans, the infection is mostly asymptomatic or a self-limiting febrile illness. However, in severe cases, liver damage, haemorrhagic illnesses, encephalitis and neurological disorders have been reported [21]. Although miscarriages in pregnant women have been attributed to RVFV outbreaks in endemic areas, data supporting such clinical outcomes is currently lacking [22]. 

Following the first isolation of RVFV in 1930s, multiple RVF epizootic outbreaks have been reported. The outbreaks were initially limited to Kenya and later spread across the eastern parts of Africa. Over the past few decades, the disease was reported in most African countries, Saudi Arabia, Yemen and serological evidence was obtained from Turkey suggesting its spread to a wider geographical area [3,23,24]. Climate change has increased the average temperatures of several countries, providing a favourable condition for more extended vector breeding. Livestock trade and international travel have introduced the disease to previously unreported territories. The possibility of aerosol transmission poses RVFV as a high-risk pathogen [25]. The geographical expansion of affected areas, along with the lack of a highly efficient and safe vaccine, underscores the unmet need for novel intervention strategies. Obtaining a better understanding of the pathogenesis of RVFV infections and host–virus interaction may aid the development of novel prophylactic and therapeutic approaches. This review aims to provide an outlook on the host immune response against RVFV. We briefly review the characteristic features of RVFV, the replication cycle and host susceptibility. We also discuss the recent advancements in understanding the innate and adaptive immune responses to RVFV infection. Finally, we discuss vaccines that were used in the past and the ones that are being developed for future use against the virus.

## 2. Virus Biology

The RVF virion is approximately 100 nm in size and has an envelope decorated with surface glycoproteins. CryoEM reconstruction reveals a quasisymmetrical, icosahedral lattice with pentameric and hexameric glycoprotein capsomers [26]. Characteristic for all bunyaviruses, the negative-sense RNA genome consists of three gene segments, S, M and L, encoding several structural and non-structural proteins. The genomic S RNA codes for the nucleocapsid (N) protein and the anti-genomic RNA encodes non-structural NSs protein [27]. The M segment encodes the glycoprotein precursor Gc and Gn, the non-structural protein NSm and accessory proteins P78, P14 and P13. The L segment encodes the viral RNA dependent RNA polymerase (RdRp). Each genome segment is flanked by complementary 3′ and 5′ untranslated regions (UTRs), thereby forming a panhandle structure. The N protein is a hexameric unit with the N terminal end enabling oligomerization and a concave cleft in the centre for binding to viral RNA. The N protein-bound RNA is thus packaged into the virion, protected from RNA-degrading enzymes and innate immune sensors of the host [27,28,29]. The viral genome segments are encapsulated by the N protein along the length of genome and the RdRp at the UTRs, creating a ribonucleoprotein (RNP) complex, essential for replication and transcription. The structure and amino acid sequence of RdRp is highly conserved among the Bunyaviridae [30]. As a result, both experimental evidence and in silico modelling suggest that the viral RdRp is an ideal target for antiviral drugs because mutations in the conserved motifs that would confer drug resistance have proven to attenuate the virus [31,32]. The M segment contains five distinct sites for translation initiation, resulting in the synthesis of different polyproteins. The signal peptides influence the translocation into organelles, differential cleaving and processing of the polyproteins to generate functional proteins [33]. The accessory protein P78, formed by the translation from first AUG, is a glycoprotein localizing to the Golgi complex and playing a role in the dissemination of virus particles in mosquito models [33]. The glycoprotein precursor molecule could be generated from all translation initiation sites and is post translationally cleaved to form the Gn spike protein and Gc class II fusion protein. Gn glycoprotein has head and stem domains, with the head domain exposed on the capsomer surface [34]. The Gc protein contains three domains mainly comprised of β-sheets, with domain II functioning as the fusion loop [35]. The structural biology approach predicts that the Gn protein forms a ring structure in pentamers, thereby shielding the Gc fusion loop. Functionally, the location of Gn on Gc might be relevant for the stepwise entry of the virus into host cells, in which the interaction of Gn with cellular entry factors would trigger endocytosis, and a low pH within endosomes enables a Gc-mediated virus-membrane fusion [36]. The non-structural proteins NSm and NSs contribute to the virulence of RVFV. NSs proteins form amyloid filaments within the infected cells which aggregate into fibrils, leading to neurological symptoms in mice [37]. The protein also downregulates host factors PKR and Abl2 to enable viral translation, IFN-β suppression, as well as actin filament formation, cell migration and spreading [38]. Protein NSm is reported to be localized on the mitochondrial membrane and prevents apoptosis of the virus-infected cells [39]. NSm protein is responsible for virus replication and dissemination from the midgut of mosquitoes and its deletion impairs virus replication kinetics [40]. The structural and non-structural proteins thus facilitate host interaction, virus replication and mediate immune evasion. 

## 3. Replication Cycle

The replication cycle starts with the attachment of the virus to the cell surface and subsequent entry into the host cell. The interaction of bunyaviruses with molecules such as DC-SIGN, L-SIGN, heparan sulphates and nonmuscle myosin heavy-chain IIA (NMMHC-IIA) expressed on the host cell surface have been reported and are likely to act as viral entry receptors [41,42,43,44,45]. More recently, low-density lipoprotein receptor-related protein 1 (Lrp1) was shown to be essential for RVFV infection [46]. Virus-receptor interaction triggers receptor clustering and subsequent endocytosis at a low pH, as observed with fluorescently tagged Uukuniemi virus (UUKV) [44]. Although caveolin-mediated endocytosis was proven to be utilized for virus entry, the role of macropinocytosis in RVFV entry cannot be ruled out [47,48]. Following endosomal uptake, the viral genome is released into the cytoplasm after pH-dependent membrane fusion, as was shown for Hantaan virus, Andes virus and RVFV [49,50]. For other bunyaviruses, like Bunyamwera virus (BUNV) and Hazara virus (HAZV), it was suggested that cellular cholesterol levels and an accumulation of potassium within endosomes play a role in membrane fusion, although it is not known whether RVFV is exploiting the same mechanism [51]. Post membrane fusion, the genome is released into the host cytoplasm and replication proceeds using the prime and realign mechanism by the RdRp and complementary RNA. Transcription is known to be initiated using cellular-capped RNA as primers, while translation is still not understood [52]. The proteins Gn-Gc heterodimerize in the endoplasmic reticulum (ER) lumen and are transported to the Golgi for post-translational modifications. Severe fever with thrombocytopenia syndrome virus (SFTSV), belonging to Phenuiviridae, was shown to upregulate the levels of autophagy-associated protein LC3-II and has been shown to exploit the autophagy machinery for the assembly and egress of nascent virion from host cells [53]. Treatment of RVFV-infected cells with sorafenib, a Raf kinase inhibitor, traps RVFV virion in the ER, hinders cellular secretory pathway thereby reducing viral egress [54]. Bunyaviruses exploit various mechanisms for their infection and replication, which is determined by host factors and viral determinants. 

## 4. Host Susceptibility

RVFV infections can have high case fatality rates in certain species. Among ruminants, sheep and goat species are highly susceptible to RVFV infections and are fatal in up to 30% of young adult animals and in over 95% in neonatal and unborn animals [55]. Cattle and camels are less susceptible to infection. Wild animals have been found to be seropositive for RVFV antibodies, for instance 31.8% of wild ruminants surveyed in Kenya during the epizootic period in 2006–2007 were positive for anti-RVFV antibodies [11]. Another critical factor determining the susceptibility and clinical manifestation of the disease is the route of infection. For instance, intranasal administration of RVFV results in higher viremia and central nervous system (CNS) infections [56]. 

RVFV infection of sheep is characterized by necrosis in the liver, gall bladder, spleen, kidney, adrenal gland and gastrointestinal tract. Necrotic foci in the liver is unevenly distributed and often extends to the periportal zone, as seen in ruminants and humans [57,58]. Apoptotic features such as cell dissociation, hypereosinophilic cytoplasm, pyknosis, karyorrhexis, etc. are also observed in injured hepatocytes, with surrounding cells exhibiting vesicular degeneration. Lymphocytic infiltration is visible in liver and lungs, while lymphocytolysis is observed in lymphoid organs. Haemorrhage and oedema are seen in the spleen and lungs of infected sheep. Inclusion bodies were observed in the liver of neonatal sheep and goats. The gall bladder had noticeable necrosis, haemorrhage and oedema and neutrophil infiltration [57,58,59]. Humans present symptoms of ocular disease and encephalitis in addition to febrile disease and haemorrhage [60]. Pathological manifestations of the CNS including encephalitis is observed in humans and experimentally infected non-human primate and rodent models. A histopathological examination of infected human brains demonstrated encephalitis with an infiltration of lymphocytes and necrosis in neurons [61]. In pregnant sheep infected with RVFV during the gestation period, viral antigens were detected in trophoblasts, syncytial epithelium and endothelium of umbilical cord. This indicates that foetal infection takes place via the haemophagus zone during maternal blood phagocytosis or infecting foetal endothelium through the maternal epithelium and trophoblasts [19]. In contrast to the vertical transmission observed in ruminants, maternal–foetus transmission in humans has not been observed. Cells of the human trophoblast cell lines A3 and Jar could be infected with wild type and NSs-deficient virus strains. The RVFV strain 35/74 replicated efficiently in cytotrophoblasts of human placental explants, suggesting that transmission from human maternal trophoblasts to foetal chorionic villi via blood is possible [19,62]. Thus, RVFV can affect several organs including the gastrointestinal system, lungs, kidney, spleen, brain and/or the placenta of susceptible hosts. 

## 5. RVFV Infection—Host Immunity and Evasion Strategies

### 5.1. Immune Evasion Mechanisms of RVFV

Virus invasion in the host is recognized by the intrinsic mechanism in cells prompting the immune system to launch a response against it. The host’s innate immune system has the capability of recognizing foreign molecules and pathogens based on common conserved features. Such molecular signatures known as pathogen-associated molecular patterns (PAMP), comprises of certain cell membrane components and genetic material of pathogens and are recognized by membrane-bound and intracellular pattern-recognition receptors (PRRs) of the host cells. Toll-like receptors (TLRs), RIG-I-like receptors (RLRs), C-type lectin receptors (CLRs), NOD-like receptors (NLRs) and AIM2-like receptors (ALRs) are some of the PRRs which recognize PAMPs and lead to the activation of the signalling cascade that acts to control invading pathogens. The N terminal domain of TLRs recognizes molecular patterns, while the C terminal domain is involved in signal transduction via the MyD88 pathway or the TRIF pathway [63]. TLRs 3, 7, 8, 9 recognize genetic material and are able to launch immune responses against these for pathogen clearance. The retinoic acid-inducible gene-I (RIG-I) identifies uncapped 5′-triphosphate ssRNA and short dsRNA formed as a result of viral replication, while the melanoma differentiation-associated gene 5 (MDA5) recognizes long dsRNA, thereby playing a major role in distinguishing virus and host cell genetic material. The binding of viral RNA with RLRs results in mitochondrial antiviral signalling protein (MAVS) recruitment, TBK-1 phosphorylation or NF-κB activation, which acts as a transcription factor for the interferons (IFN) and interferon-stimulated genes (ISG) [64]. IFNs interact with the IFN receptor in autocrine or paracrine manner, thereby phosphorylating STAT protein, leading to the transcriptional activation of interferon-stimulated genes under the control of an interferon-stimulated response element or gamma interferon activated sequence [65]. 

The RVFV genome is recognized by the RIG-I molecule and activates the downstream signalling process, including the production of interferon-inducible transmembrane proteins (IFITM) -2 and -3, which restrict the virus life cycle in early stages post internalization and prior to replication in vitro [66,67]. As an immune evasion mechanism, RVFV modifies 5′ termini and escapes RIG-I recognition [68]. Protein atypical RIO Kinase 3 (RIOK3) is an essential regulator of the IFN-β pathway and NF-κB pathway. The constitutively expressed form is crucial in IFN-β production, while the alternative splice variant is involved in NF-κB generation [69,70]. RVFV triggers inflammasomes as indicated by the interaction of NLRP3 and MAVS in infected cells [71]. RVFV infection in liver cells led to the NSs protein-dependent generation of reactive oxygen species, p65 production and triggering the NF-κB pathway [72]. The NSs proteins in the RVFV MP12 strain activate the ATM DNA damage pathway, leading to cell cycle arrest at the S phase, while NSs of ZH548 arrest the cell cycle at the G_0_/G_1_ phase. The cell cycle arrest halts host cellular processes, thereby favouring virus replication [73,74]. The exploitation of the DNA damage pathway is also utilised by other viruses for facilitating replication, including human parvovirus B19, Epstein–Barr virus, Porcine epidemic diarrhoea virus, SFTSV, etc. [75,76,77,78]. Similarly, the abrogation of interferon signalling which is critical for antiviral immunity is observed in related phleboviruses. SFTSV infection is highly reduced upon IFN-γ treatment prior or post infection. The IFN-γ signalling is disrupted by NSs proteins through sequestration and downregulation of STAT1 expression [79]. NSs proteins of sandfly fever viruses prevent the phosphorylation of Jak-1 and Stat-1 and inhibit the JAK-STAT pathway for the production of ISG [80]. In RVFV, the NSs protein interacts with the host Sin3A-associated protein 30 (SAP30) component of the histone deacetylase complex, thereby maintaining the IFN-β signalling in a transcriptionally silent state and promoting virus replication. NSs mutants with amino acid deletions which are essential for SAP30 binding abrogated its interaction with the IFN-β promoter sequence and enabled survival of the infected cells [81]. The host protein PKR has a promoter region activated by IFN that is responsible for activating immune responses in paracrine signalling. The NSs degrades double-stranded RNA (dsRNA)-dependent protein kinase (PKR), thereby ensuring that innate immunity is blocked in uninfected cells. PKR is also responsible for phosphorylating eukaryotic initiation factor 2α (eIF2α) to enable efficient host mRNA translation, which prevents viral translation. PKR degradation by NSs also functions to prevent the phosphorylation of eIF2α and ensures viral translation and shutdown of the host’s immune defence [82]. Incoming RVFV virions trigger canonical autophagy pathways in several cell lines. RVFV infection induces autophagy in mouse embryonic fibroblasts, primary mouse hepatocytes, primary rat mixed neuroglial cultures and human osteosarcoma cell lines (U2OS). Moreover, treatment of these cell lines with autophagy-activating drugs prior to infection showed antiviral activity against RVFV [83]. In contrast, RVFV infection in differentiated macrophage cells (THP-1^PMA^) resulted in the interaction of viral nucleoprotein and host sequestome 1 (SQSTM1), triggering autophagosome formation and aiding viral replication. However, in Huh7 cells or human umbilical vein endothelial cells (HUVEC), RVFV infection triggered autophagy, which was dispensable for virus replication [84]. Hence, RVFV-triggered autophagy has a pro-viral or anti-viral role, depending on the cell type infected. Exosomes are a category of extracellular vesicles formed as a result of endocytosis and subsequent fusion with the cell membrane [85]. Exosomes formed as a result of RVFV infection containing viral RNA could activate the RIG-I pathway, leading to Interferon-β (IFN-β) production and upregulating autophagic flux [86]. Thus, various intrinsic cellular defence pathways can influence the innate immune response to infection (Figure 1).

### 5.2. Innate Immune Response to RVFV Infection

RVFV infections induce strong cytokine and chemokine responses in target organs, which are not only critical for the recruitment of innate immune cells to the sites of infection but are also crucial for priming an effective adaptive immune response. Cytokine responses to RVFV infection have been sparsely studied in goats, sheep and cattle. Goats infected with the ZH501 strain had negligible levels of type I IFN production, due to the suppression of interferon production by the NSs gene. Nevertheless, IL-12 and IFN-γ levels were higher in infected goats compared to uninfected goats. Pro-inflammatory cytokines TNF-α, IL-6, IL-1β expression increased two days post infection (dpi) and contributed to the virus clearance from infected goats [87]. 

Human monocyte-derived macrophages (MDM) could be productively infected in vitro with wild type ZH501 and NSs-deficient (ΔNSs) strains of RVFV. As opposed to the wild type virus, the ΔNSs strain induced pro-inflammatory cytokines IFN-α2, IFN-β and TNF-α which restricted virus replication and cytopathic effect (CPE) in human MDM. [88]. The infection of murine bone marrow-derived macrophage (BMDM) with RVFV strains ZH501 or MP12 resulted in a lower expression of pro-inflammatory cytokines IFN-β, IFN-γ, TNF-α than after infection with recombinant MP12 with an in-frame deletion of 69% of the NSs gene (rMP12-C13). In general, the production of inflammatory cytokines was downregulated in ZH501-infected cells, while for other cytokines such as IL-3, IL-4, IL-10 and IL-17, the expression was comparable to rMP12-C13 and MP-12 infected cells. The diminished cytokine response to wild type virus can be mostly attributed to the immune suppressive capabilities of NSs [89]. Similar observations were also made in the human choriocarcinoma cell line Jar and the trophoblast cell line A3, in which inflammatory cytokines such as TNF-α, IL-6 and IL-8 were highly expressed [62]. Serum samples collected from infected humans during the outbreak in 2010–2011 in South Africa were tested for cytokine levels. Higher level of the anti-inflammatory IL-10 was observed in fatal cases than in the survivors and negative control subjects [90]. The level of pro-inflammatory cytokines IL-6 and IL-8 was elevated in infected subjects and correlated with liver damage, while levels of CCL5/RANTES responsible for attracting immune cells were reduced in fatal cases and correlated with haemorrhage. CXCL-9, CXCL-10 and MCP-1 levels were higher in all infected samples, compared to the control group. Thus, the balance between anti- and pro-inflammatory cytokines and chemokines largely defines the disease outcome of the infection [90]. 

In experimental mouse models, the genetic background of the mouse strains used determines the response to and the outcome of RVFV infection. MBT/Pas mice displayed a higher susceptibility to RVFV infection than BALB/cByJ mice, which is surprising because MBT/Pas mice had relatively high serum IFN-α levels. However, their IFNAR1 receptor expression on leukocytes was lower, which abolished the protective effect of type I IFN [91,92]. Interestingly, the site of RVFV infection is important in determining the disease manifestations. Intranasally infected mice were found to be at higher risk of high viral loads in the brain and subsequent development of encephalitis than mice infected by injection in the footpad. Also, the extent of spreading to peripheral organs was lower. The infiltration of leukocytes, the production of pro-inflammatory cytokines and a lack of virus clearance were the leading factors associated with encephalitis in mice [93].

The pathogenesis and response to RVFV infection by aerosols was also studied in non-human primates (NHP), namely African green monkeys (AGM), rhesus macaques, cynomolgus macaques and common marmosets [94]. African green monkeys and marmosets showed signs typical for RVFV, including anorexia, weight loss and neurological distress at 9–12 dpi [94]. Lethally infected monkeys had high virus loads in the CNS, while peripheral organs had a lower viral load. The delayed increase in monocytes, dendritic cells and the pro-inflammatory cytokines IFN-γ, IL-8, sCD40L in the blood of lethally infected monkeys results in an inefficient immune response and subsequent morbidity. A sublethal infection led to the upregulation of inflammatory cytokines within 3 dpi, providing protection against severe disease [95]. Thus, the timing of innate response induction is crucial in determining the survival of infected organisms [95]. 

### 5.3. Adaptive Immune Response

Sero-epidemiological studies conducted during epidemic or inter-epizootic phases in RVFV endemic regions showed that humans and ruminants have RVFV-specific antibodies [96,97,98]. In humans naturally exposed to RVFV, high titers of virus neutralizing (VN) antibodies directed against the glycoproteins were detected, in addition to low frequencies of N-, Gc- and Gn-specific T cells [99]. Virus-specific cellular and antibody responses were also observed in goats twenty-one days after experimental infection with the RVFV-ZH501 strain [87]. Also in sheep, virus-neutralizing antibody responses were observed upon RVFV infection [100]. Serum samples obtained from cattle, goats and sheep collected two months after a RVFV outbreak contained virus-specific antibodies, as demonstrated by ELISA. In this study, 65% of RT-PCR positive sera were still seronegative, suggesting that these animals were acutely infected during the outbreak [101]. Finally, non-human primates experimentally infected with RVFV developed an antibody response [94]. Thus, in various susceptible species such as ruminants, humans, wild ungulates and camels, RVFV infection induces virus-specific antibody responses [11,102,103,104]. A sublethal dose of RVFV administered in African green monkeys also led to the expansion of both cytotoxic and helper T lymphocytes in peripheral blood [95]. The induction of long-lasting, robust antibody- and cell-mediated immunity affords protection against subsequent infection in previously infected hosts. 

Studying adaptive immune responses to wild type strains in the experimental infection of rodent models is often challenging due to rapid disease progression and high mortality within a few days after infection. Moreover, the adaptive immunity to wild type RVFV infection is not highly explored, compared to the response induced by viral proteins or attenuated strains. Hence, humoral and cellular responses to the respective RVFV viral proteins have been characterized using expression systems or recombinant protein. One of the first studies on RVFV-induced humoral response utilized baculovirus vector expressing glycoproteins for immunizing mice. The immunized mice developed an antibody response against the glycoproteins, as evident by the presence of antibodies and the survival of passively immunized mice [105]. Rabbits immunized with Gn glycoprotein mounted a specific antibody response. Their B cells were used to generate hybridoma cells that produced VN antibodies which afforded protection against the challenge infection in naïve recipient mice [106]. The immunization of sheep with an expression plasmid encoding the ectodomain of Gn elicited an antibody response which afforded protection, as judged by an accelerated viral clearance and reduced disease severity [107]. In naturally infected humans, Gn but not Gc glycoprotein was the major viral protein inducing a sustained VN-antibody response, while both Gn and Gc induced T cell responses [99]. The immunization of sheep with a recombinant Equine herpesvirus-1 expressing the RVFV Gn and Gc proteins elicited a serum VN-antibody response [108]. The immunization of mice with recombinant nucleocapsid protein along with Alhydrogel induced an N-specific Th2 response [109]. 

The importance of adaptive immunity in controlling brain pathology was studied in C57BL/6J mice depleted of B cells, CD4^+^ T cells or CD8^+^ T cells using antibodies by infecting subcutaneously with ∆NSs strain of virus. B cell-deficient mice succumbed to the challenge at 21 dpi, indicating the importance of B lymphocytes in reducing virus infection [110]. Surprisingly, CD8^+^ T cells were not essential for protective immunity as the CD8^+^ T cell-depleted mice did not succumb to disease after challenge infection. CD4^+^ T cell depletion resulted in high virus RNA copy numbers in the liver and brain and reduced antibody titer compared to the mock-depleted mice. CD4^+^ T cell-depleted mice had a higher viral load and pro-inflammatory cytokine levels in the brain, including CCL2, CCL4, CCL5, CCL7, CCL12, CXCL1, CXCL10, CXCL-13, TNF-α and IL-1β [110]. A recent study also confirmed that the NSs-deleted RVFV strain is capable of causing encephalitis in the absence of CD4^+^ T cells working synergistically with monocytes and CD8^+^ T cells [111]. Moreover, NSG mice deficient in T-, B-, natural killer (NK)-cells and macrophage-mediated immunity had viral replication in organs including the brain, with one animal developing significant clinical signs [112]. These observations lead to the conclusion that CD4^+^ T cells control progression to late onset encephalitis in mice.

## 6. RVFV Vaccine Development

Since the discovery of RVFV as a causative agent of disease in animals and humans, several attempts have been made to develop safe and effective vaccines. One of the first vaccine candidates tested was based on the Smithburn strain, which was accidentally isolated while studying Yellow Fever in Uganda by infecting mice intracerebrally with mosquito extracts. Later, it was found that some of these mosquitoes were vectors for RVFV, and the disease pathology of a few infected mice were similar to RVF manifestations [113]. Follow-up studies on the Smithburn strain suggested that the lower number of passages leads to a pantropic virus, affecting the liver predominantly, and higher passages form a neurotropic virus against which the pantropic virus induced higher antibody titers and protection [114]. Alpacas in South Africa were vaccinated with the Smithburn vaccine as a precautionary measure against RVFV, although it was previously unreported in the species. A few of the vaccinated individuals died and histopathological examination revealed meningoencephalitis and RVFV antigens in the brain. A sequence analysis confirmed the identity of the Smithburn strain isolated from the cerebrum of the diseased alpacas and excluded natural RVFV infection as the cause of death [115]. 

### 6.1. Formalin-Inactivated Vaccines

Vaccine candidates NDBR-103 and TSI-GSD-200 are formalin-inactivated virus preparations which were administered previously in high-risk groups. Even though a virus-neutralizing antibody titer was generated post vaccination, few individuals that received NDBR-103 developed Guillain-Barré Syndrome and the safety of the vaccine candidate was questioned subsequently [116,117]. Individuals working in high-risk conditions with RVFV vaccinated with TSI-GSD-200 showed a robust humoral response with sustained antibody titers. Studies on the longevity of TSI-GSD-200-induced antibody responses showed that antibody titers waned 6–12 months post vaccination, which could be overcome by booster vaccinations. The major disadvantage of the requirement of booster vaccinations is the inability to induce immediate protective antibody responses during outbreaks of RVFV infections [118,119]. Nevertheless, RVFV-specific T cell responses were detectable in immunized individuals, even 24 years post vaccination [120]. 

### 6.2. MP12 

The RVFV strain MP12 was developed from the non-fatal ZH548 strain, which was passaged in suckling mouse brains, propagated in foetal rhesus lungs and human diploid fibroblasts and later mutated through alternate passaging in the absence and presence of 5-fluorouracil [121]. The MP12 vaccine strain differs from the parent virus strain ZH548 by 25 mutations, rendering it highly attenuated and temperature-sensitive [122,123]. Numerous studies have been conducted on the MP12 strain to analyse its efficacy, safety, threat of transmission or reversion to virulence. The vaccine was found to be safe in vaccinated sheep with no adverse effects and no potential transmission to mosquito vectors of four different species. MP12-induced antibodies persisted even 24 months post vaccination [124]. A single-dose MP12 vaccination was safe in alpacas and elicited an antibody response [125]. An intramuscular injection of rhesus macaques with the MP12 strain did not cause clinical signs and induced virus-specific serum IgM and IgG antibody responses, which persisted even up to 126 days post vaccination [126]. The safety of the vaccine in pregnant ewes and lambs was tested by inoculating ewes at their second trimester and neonatal lambs. The administration of MP12 did not cause lesions in peripheral organs and induced VN-antibody responses. Neonatal lambs inoculated with MP12 had virus present in the liver, spleen and kidney up to day 4 post inoculation, albeit at low titers. Newborn lambs of vaccinated ewes lacked MP-12 antibodies in peripheral blood at birth, prior to colostrum feeding. Colostrum feeding introduced maternal antibodies in the lambs, indicating acquired immunity through nursing [123]. The immunization of sheep with MP12 during the early stages of pregnancy has caused adverse effects including foetal malformation and teratogenicity [127]. Also, in four-month-old calves, the injection of MP12 caused adverse effects, including bronchopneumonia in five of ten vaccinated animals, and lesions in the liver, lung, gall bladder and trachea up to 10 days post inoculation. Viral antigen could also be detected from the liver of a vaccinated calf [128]. The efficacy of the vaccine was demonstrated in macaques by intravenous or aerosol challenge infection with the strain ZH-501 [126,129]. MP12 was also tested as a vaccine candidate in phase 1 and 2 clinical trials. Also, in human study subjects, MP12 proved to be immunogenic and induced VN antibodies, which persisted up to 5 years. [130,131].

### 6.3. Clone 13

Clone 13 is a live attenuated RVFV strain derived from the 74HB59 strain by plaque purification. The clone 13 has a 69% deletion in-frame within the NSs gene, with intact N- and C-terminals, and contains nucleotide substitutions in the N and NSs gene [132]. Clone 13 was evaluated for its safety and efficacy against virus challenge in susceptible species such as sheep, cattle and goats. In sheep, the administration of clone 13 was relatively safe as no abortions were reported in pregnant ewes in the early or later stages of pregnancy. The vaccinated sheep developed VN-antibody responses and were protected from challenge infection [133]. Also, in seven-month-old calves, vaccination with clone 13 induced protective immunity [134]. Field studies conducted in Senegal and Tanzania in herds of ruminants confirmed the safety and immunogenicity of clone 13 [135,136]. However, adverse effects including abortions, stillbirths, and malformations have been reported in ewes vaccinated at 50 days of gestation, indicating the vertical transmission of clone 13 during early stages of pregnancy [137]. Thus, although clone 13 proved to be efficacious, its safety especially in pregnant ruminants is still a matter of debate. 

### 6.4. RVFV-4s 

More recently, live attenuated RVFV vaccines were developed with a genome consisting of four gene segments (RVFV-4s) by splitting the Gn and Gc segments and deleting the NSs gene [138]. Two versions of the vaccine candidate were designed: a vaccine for veterinary use with RVFV strain-35/74 as backbone, and a vaccine for humans with Clone 13 as the backbone [139]. Vaccination with RVFV-4s induced VN antibodies primarily against the Gn proteins and afforded young lambs protection against challenge infection with the wild type virus [140]. The safety of the vaccine was tested by administering RVFV-4s in pregnant ewes at 50 days of gestation. The vaccine induced VN antibodies and did not result in infection in foetuses or ewes [141]. Additionally, the veterinary RVFV-4s vaccine candidate did not disseminate, shed or spread into the environment in a study cohort and did not revert to a virulent strain when lymph node suspensions of immunized lambs were administered to naïve lambs. A single vaccine dose was sufficient to protect lambs, goat kids and calves from virus infection [142]. Both the human and veterinary RVFV-4s vaccine candidates were tested in common marmosets and induced VN-antibody responses [139]. The RVFV-4s vaccine is highly efficacious and safe in all tested species and is a promising candidate for a veterinary and human RVF vaccine. 

### 6.5. DNA Vaccines 

Various other vaccine candidates have been developed by modifying virus strains, the deletion of the NSs and NSm genes and producing expression plasmids encoding RVFV viral proteins [143,144]. A DNA vaccine encoding the Gn and Gc glycoproteins (pWRG7077-RVFV_-NSm_) was highly immunogenic in BALB/c mice. Moreover, the challenge of three-time immunized mice with the ZH501 strain enabled survival, compared to the control mice [145]. A plasmid vector consisting of the Gn gene coupled to the murine complement protein C3d (Gn-C3d) elicited a strong immune response in mice and was effective in protecting mice against ZH501 challenge. The passive immunization of naïve mice with sera from vaccinated mice one hour prior to infection protected these against ZH501 challenge [146].

### 6.6. Viral Vectored Vaccines 

Viral vector systems have also been explored as RVFV vaccine candidates. A recombinant Venezuelan Equine Encephalitis Virus (VEEV) expressing Gn proteins fused to the E2 protein on the cell surface was immunogenic in mice after a single vaccination and afforded protection against challenge infection with the RVFV ZH501 strain [147]. Modified Vaccinia Virus Ankara (MVA) have previously been utilized as vaccine candidates against human immunodeficiency virus, Japanese encephalitis virus, *P. falciparum*, *M. tuberculosis* and other pathogens [148,149,150]. rMVA-expressing RVFV glycoproteins or nucleoprotein provided sterile protective immunity in mice against challenge infection with RVFV. Protective immunity was a cumulative effect of RVFV-specific antibody and T cell response [151]. Vaccination with Lumpy skin disease virus (LSDV) recombinantly expressing RVFV glycoprotein induced protective immunity in Merino sheep against RVFV challenge infection [152]. A chimpanzee adenovirus vectored vaccine consisting of Gn and Gc glycoproteins with adjuvants Matrix-M™ and AddaVax™ elicited strong humoral immunity, even at eight weeks post vaccination, and a CD8+ T cell response two weeks post vaccination in mice [153]. The safety and efficacy of the vaccine candidate was established in pregnant ewes, while low level viremia were detected in placenta and plasma of pregnant does post challenge [154]. Recently, a novel method was developed with bacterial outer membrane vesicles, decorated with RVFV Gn head domain. Immunization in mice induced IgG antibodies and antigen-specific T cell responses [155]. Thus, numerous strategies and platforms to develop RVFV vaccines are under investigation to test their safety and efficacy, as summarized in Table 1.

## 7. Conclusions

Rift Valley Fever is a continuing threat to livestock and human health. Global warming could provide a fertile breeding ground for mosquitoes and interaction between humans and animals might increase the risk of transmission of the virus [158]. Sero-epidemiological evidence indicates that endemic regions are expanding, which underscores the need for close surveillance and vigilance. Considering the losses in agriculture and the public health threat, a one health approach to combat this virus seems warranted and the availability of safe and efficacious vaccines for veterinary and medical use is urgently needed and, therefore, RVFV is on the WHO R&D blueprint list [159].

Despite the fact that RVFV was already isolated in the early 1930s, our understanding of its epidemiology, pathogenesis and immunity is limited. The inter epizootic maintenance of virus and reservoir hosts are not entirely known. A recent study implicated black rats as reservoirs of RVFV because they could be infected persistently for one month without disease manifestations [160]. A RT-PCR analysis of the bat species *Pipistrellus deserti* in Egypt identified partial sequences of the virus, suggesting its role as a potential reservoir [161]. Further studies focusing on interspecies interaction and virus prevalence could improve our understanding of the RVFV transmission cycle and inter-epidemic virus maintenance. A phylogenetic characterization of virus isolates indicated that distinct lineages are highly similar in amino acid sequences. However, virus replication rates and mortality rates of different strains are highly variable [162]. Insights into the nature of genome variation and the impact on virus pathogenesis would be of interest. The high sequence similarity is also advantageous for the development of targeted therapeutics, as antivirals or vaccines developed against one strain would be effective against other RVFV strains.

RVFV replication and evasion of innate host immune responses is facilitated by the NSs protein, which inhibits interferon response and is an important virulence factor. Induction of the interferon response is crucial for the innate and adaptive immune response, even though the interferon response is unable to control virus replication in susceptible hosts directly. Moreover, the severity of the disease also depends on the genetics of the organism, for instance, different strains of mice respond distinctly to infection, where some are more susceptible to lethal infections [163]. The clinical outcome of infection also depends on the age of host, with foetuses and younger organisms being more susceptible to infection and prone to fatal outcomes, and time of induction of immune response. In the absence of a functional NSs protein, the IFN type I response of the host is efficient in controlling (mutant) virus replication to a certain extent and preventing the development of severe disease. Furthermore, a balance between pro- and anti-inflammatory cytokine levels determines disease severity. During natural infections with wild type RVFV, an efficient pro-inflammatory cytokine response may not be induced, resulting in a more severe disease. Sustained virus-specific antibody- and cell-mediated immune responses afford protection against subsequent infections and should be the target for vaccination strategies. 

Various vaccine candidates have been developed to protect susceptible hosts from RVFV infections. Formalin-inactivated vaccines caused side effects such as teratogenicity and neurological complications, while some live attenuated vaccines were under-attenuated or pose a risk for reversion to virulence. Other vaccine candidates like subunit vaccines and virus-like particles are still in the early stages of development.

Newer and more promising vaccine candidates have been designed using segmentation of the viral genome and complete deletion of non-structural NSs and/or NSm genes. These vaccine candidates proved to be highly immunogenic, have an excellent safety profile and are currently under investigation in clinical trials. To obtain a better understanding of the mode of action and correlates of vaccine-induced protection, further research is warranted. The use of highly effective vaccines in livestock and humans in endemic regions may mitigate the impact RVFV has on agriculture and public health as an effective “One health” approach.

## Figures and Tables

**Figure 1 pathogens-12-01174-f001:**
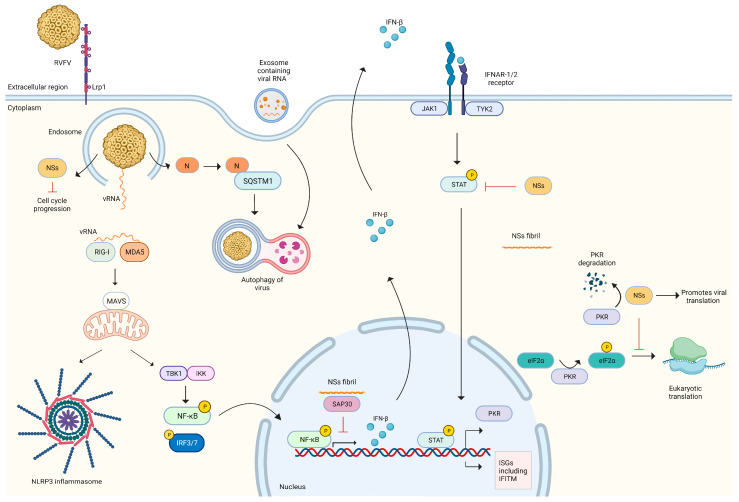
Immunity to RVFV infection in host cells and evasion strategies adopted by the virus. RVFV uses Lrp1 as one of the attachment factors and enters host cell by endocytosis. Low pH in endosome triggers membrane fusion, release of virus genome and proteins. Viral RNA is sensed by RIG-I/MDA5 which interacts with MAVS and activates TBK1 signalling. NF-κB is phosphorylated and translocates to the nucleus, resulting in IFN production. MAVS also triggers NLRP3 inflammasome formation. IFN interacts with IFNAR receptor in autocrine or paracrine manner, phosphorylating STAT protein, which acts as transcription factor for ISGs and PKR. PKR phosphorylates eIF2α, thereby initiating eukaryotic translation. Viral proteins released from endosome including nucleoprotein and NSs affect host immune response. N protein interacts with SQSTM1 to generate autophagosome. Exosomes originating from RVFV-infected cells can also trigger autophagy. NSs protein suppresses innate response by several methods, including cell cycle arrest and inhibiting STAT phosphorylation, preventing IFNAR signalling. NSs forms fibrils which localize in nucleus, interact with SAP30 and prevent IFN transcription. NSs also degrades PKR, halting eukaryotic translation and promoting virus RNA translation. Created with BioRender.com (accessed on 20 August 2023).

**Table 1 pathogens-12-01174-t001:** Representative RVFV vaccine candidates developed over the years.

Vaccine Candidate	Type of Vaccine	Immune Response	Species Tested	Comments	References
Smithburn	Live-attenuated	Neutralizing antibody	Cow, goat, alpaca	Adverse effects: abortion, pathologic manifestation	[134,156,157]
NDBR-103	Formalin inactivated	Neutralizing antibody	Humans	Side effect: reported case of Guillain-Barré Syndrome	[116,117]
TSI-GSD-200	Formalin inactivated	Neutralizing antibody, T cell	Humans	Booster vaccinations required	[118,119,120]
MP-12	Live-attenuated	Neutralizing antibody	Sheep, goat, alpaca, rhesus macaque, humans	Adverse effect for foetus when administered in early pregnancy, phase 1 and 2 clinical trials	[123,124,125,126,128,129,130,131]
Clone-13	Live-attenuated	Neutralizing antibody	Sheep, goat, cattle	Vertical transmission	[133,134,135,136,137]
RVFV-4s	Live-attenuated	Neutralizing antibody	Sheep, goat, cow, common marmoset	No evidence of vertical transmission	[139,140,141,142]
arMP-12-ΔNSm21/384 ^1^	Live-attenuated	Neutralizing antibody, IFN-γ response	Sheep		[143]
rZH501-ΔNSs:GFP, rZH501-ΔNSs:GFP-ΔNSm ^2^	Live-attenuated	Neutralizing antibody	Rats		[144]
pWRG7077-RVFV_-NSm_	DNA-vectored	Neutralizing antibody	Mice		[145]
Gn-C3d	DNA-vectored	Neutralizing antibody	Mice	Protects passively immunized mice against RVFV-ZH501 challenge	[146]
RVFV-BLP ^3^ (Gn head)	Bacterium-like particle	Antibody and T cell response	Mice		[155]
rLSDV-RVFV	Virus-vectored	Neutralizing antibody	Sheep		[152]
rVEEV-RVFV	Virus-vectored	Neutralizing antibody	Mouse		[147]
rMVA-RVFV	Virus-vectored	Neutralizing antibody, T cell	Mouse		[151]
ChAdOx1-RVFV ^4^	Virus vectored	Neutralizing antibody	Mouse, sheep, goat	Viremia in placenta of pregnant doe challenged with RVFV-35/74	[153,154]

^1^ arMP-12-ΔNSm21/384- recombinant MP12 with M gene deletion from nucleotide 21 to 384. ^2^ rZH501-ΔNSs:GFP, rZH501-ΔNSs:GFP-ΔNSm- recombinant ZH501 with deletion of NSs gene or NSs and NSm genes and coupled with GFP. ^3^ BLP- bacterium like particle. ^4^ ChAdOx1-chimpanzee adenovirus vector.

## Data Availability

Not applicable.

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
