# Peer review of "Rift Valley Fever Virus—Infection, Pathogenesis and Host Immune Responses"

_pathogens, 2023, doi:10.3390/pathogens12091174_

Round 1

Reviewer 1 Report

The review article nicely summarizes the literature on pathogenesis, immunity, and vaccine efforts for Rift Valley Fever Virus (RVFV). The authors cover basic biology of RVFV, cellular targets, innate immune responses and viral proteins that antagonize the response, adaptive immunity, potential correlates of protection, and finally they discuss current vaccine approaches. The review is well written and easy to follow. The references appropriately support the text.

Minor comments:

1.       For Figure 1: RVFV exosomes induce autophagy in neighboring cells, so the arrow from the autophagosome to the exosome should be reversed (going into the cell rather than out of the cell). RIG-I typically does not signal directly to TBK1/IKK. This pathway goes through MAVS, so the arrow from RIG-I to TBK1/IKK should really be MAVS to TBK1/IKK. Additionally, IRF3/7 normally is activated as well in this pathway. In addition to RIG-I, the reference also suggests a potential redundancy with MDA5, so it should be included in the figure.

2.       On lines 281 and 282, MBT mice are mentioned. Then on line 285, MBT/Pas mice are mentioned. These are presumably the same mouse line. The MBT/Pas name should be used for clarity.

3.       The DNA vaccines highlighted in the text should be included in Table 1.

4.       Table 1: For the ChAdOx1-RVFV vaccine, the placental viral burden in the does was after challenge not from the vaccine. This should be clarified in the table.

Author Response

We highly appreciate the positive feedback given by the reviewer. We have addressed all the concerns raised by the reviewer in the revised manuscript. Below is our point by point response.

Minor comments:

  1. For Figure 1: RVFV exosomes induce autophagy in neighboring cells, so the arrow from the autophagosome to the exosome should be reversed (going into the cell rather than out of the cell). RIG-I typically does not signal directly to TBK1/IKK. This pathway goes through MAVS, so the arrow from RIG-I to TBK1/IKK should really be MAVS to TBK1/IKK. Additionally, IRF3/7 normally is activated as well in this pathway. In addition to RIG-I, the reference also suggests a potential redundancy with MDA5, so it should be included in the figure.

Response: The figure 1 has been modified to include the following details:

  1. Exosome is shown to be uptaken by the host cell and undergoes autophagy.
  2. RIG-I signaling has been modified to show the signaling through MAVS, TBK1/IKK and phosphorylation of IRF3, IRF7 and NF-κ
  3. Yes, as mentioned RIG-I and MDA5 is showing redundancy, and recognizes RVFV RNA during infection. MDA5 is also added to the figure as per suggestion.

Moreover, NSs fibril interaction with SAP30 to inhibit interferon transcription has also included in the figure.

  1. On lines 281 and 282, MBT mice are mentioned. Then on line 285, MBT/Pas mice are mentioned. These are presumably the same mouse line. The MBT/Pas name should be used for clarity.

Response: MBT and MBT/Pas mice are the same mouse line. Therefore, in lines 298 and 300, MBT was changed to MBT/Pas.

  1. The DNA vaccines highlighted in the text should be included in Table 1.

Response: The following vaccines have been included in Table1:

  1. arMP-12-ΔNSm21/384
  2. rZH501-ΔNSs:GFP and rZH501-ΔNSs:GFP-ΔNSm
  3. pWRG7077-RVFV-NSm
  4. Gn-C3d
  5. RVFV-BLP

4. Table 1: For the ChAdOx1-RVFV vaccine, the placental viral burden in the does was after challenge not from the vaccine. This should be clarified in the table.

Response: In comments section of Table 1 for ChAdOx1-RVFV vaccine, “Viremia in placenta of pregnant doe” is replaced with “Viremia in placenta of pregnant doe challenged with RVFV-35/74”.

Reviewer 2 Report

The review of Nair et al. gives a comprehensive overview of different aspects of RVFV pathogenesis, immunology and the current state of vaccines. The review is written clearly and it can be followed easily. However, I am missing a the coherent line of the review. Although all different paragraphs offer a well structured and detailed description of the chosen subtopic, it is partially unclear where this information should lead the reader to. You can generate this missing connection via a section in the introduction which describes in more detail what this review specifically aimed for and subsections can be used for.

Specific comments:

ll.32: Small detail, but camels are no ruminants

ll.33: The list of wild animals that were found to be RVFV seropositive is much longer. Indicate that with an etc. or make your list complete (not needed in my opinion).

ll.33: "In animals, RVFV infection..." - consider to remove this sentence, since it hinders the reading flow, starting with rather virus ecology and then describing the transmission cycle. Section about tissue tropism etc. is following later anyway. 

ll.39: Replace "starts" with "start"

ll.91: You already mention the Gn/Gc pentamers and hexamers in ll. 64. Consider to remove

ll.127: What is the relevance of SFTSV in this case?

ll.141: But what is the relevance of intranasal infection with RVFV?

ll.144: Insert an "and" between adrenal gland and gastrointestinal tract

Chapter 4: What about the liver? The focus of this chapter is a bit heavy on the CNS for being a rather rare observation after RVFV infections.

ll.151 The "and other experimentally infected non human primate" is misleading when you describe humans before. Remove the "other"

Chapter 5: Consider to chose another title, since you do not only describe the host immunity but also the evasion of it by the virus etc. Maybe also consider to give this first section (ll.166-232) a separate subheading (5.1).

ll.201-208: Same as questioned before, why is that mentioned here?

Figure 1: The heading is misleading cause not only immune response is visualized. Its rather viral replication that you show.

ll.281: Introduce abbreviations (MBT).

ll.293-306: How does that fit into the innate immune response chapter? Gets relevant for the subtopic from ll. 309 onward. The previous passage can be summarized in one sentence. 

ll.335: In this sentence you indicate the humoral immune responses to specific RVFV proteins is studied because infection of animal models come along with high mortality and is therefore hard to study. Do consider that an infection of relevant animals (such as small ruminants) is barely lethal and I wouldnt agree that studies you describe following upon this sentence are performed based on this rational.

ll. 354: Consider to have a look into this reference as well: doi: 10.3390/v14020350.

Table 1: The vertical transmission listed for Clone 13 is also observed for MP12 vaccinations.

Consider to move ll.516-529 after l.499, this would enhance the reading flow.

Author Response

We are grateful for the constructive feedback from the reviewer. As suggested by the reviewer, an additional section has been included in the introduction (Iines.57-62) highlighting the topics the review addresses. The section will provide readers an overview of the review and the specific information provided in the following sections. 

Specific comments:

ll.32: Small detail, but camels are no ruminants

Response: We apologize for this error and the The sentence has been rephrased as “….cattle and ungulates like camels…” in II.32.

ll.33: The list of wild animals that were found to be RVFV seropositive is much longer. Indicate that with an etc. or make your list complete (not needed in my opinion).

Response: The statement has been rephrased and Additional references pointing to serological evidence for RVFV infection has also been included.  

ll.33: "In animals, RVFV infection..." - consider to remove this sentence, since it hinders the reading flow, starting with rather virus ecology and then describing the transmission cycle. Section about tissue tropism etc. is following later anyway.

Response: This statement has been deleted in the revised manuscript (II.33.)

ll.39: Replace "starts" with "start"

Response: The grammatical error was corrected in II.38.

ll.91: You already mention the Gn/Gc pentamers and hexamers in ll. 64. Consider to remove.

Response: The repeated statement from II. 96 was removed

ll.127: What is the relevance of SFTSV in this case?

Response: SFTSV belongs to family Phenuiviridae, and is phylogenetically related to RVFV. The egress mechanism for RVFV is not entirely known. Hence, the assembly and egress mechanism of a related virus (in this case SFTSV) was quoted hinting at the likelihood of RVFV using similar mechanisms. Inhibition of RVFV egress when treated with sorafenib has been included in the text (II. 132-134) to provide information regarding involvement of secretory pathway in viral egress.

ll.141: But what is the relevance of intranasal infection with RVFV?

Response: Subcutaneous or intraperitoneal administration of RVFV in experimental animal models leads to severe hepatic lesions and hemorrhagic disease whereas intranasal infection replicates the neurological symptoms observed in humans. Hence, the intranasal route of infection has been used to study RVFV mediated encephalitis in experimental animal models.

ll.144: Insert an "and" between adrenal gland and gastrointestinal tract.

Response: An “and” was added in II.149.

Chapter 4: What about the liver? The focus of this chapter is a bit heavy on the CNS for being a rather rare observation after RVFV infections.

Response: RVFV encephalitis although rare is a major cause of concern. We agree that there should be little more focus on hepatic disease. Additional statements regarding liver pathology has been added (II. 149-153).

ll.151 The "and other experimentally infected non human primate" is misleading when you describe humans before. Remove the "other".

Response: “Other” has been removed as suggested from II.159.

Chapter 5: Consider to chose another title, since you do not only describe the host immunity but also the evasion of it by the virus etc. Maybe also consider to give this first section (ll.166-232) a separate subheading (5.1).

Response: Heading rephrased and subheading added to first section of chapter 5, II.176.

ll.201-208: Same as questioned before, why is that mentioned here?

Response: Although NSs protein in bunyaviruses act for inhibiting the interferon response, the mechanisms utilized by the viruses differ. II.201-208 (revised, II. 213-220) explains this by citing SFTSV and sandfly fever viruses as examples.

Figure 1: The heading is misleading cause not only immune response is visualized. Its rather viral replication that you show.

Response: The heading has been modified slightly as “Immunity to RVFV infection in host cells and evasion strategies adopted by the virus”. Since the replication cycle was not completely represented in the figure, this term has been avoided. 

ll.281: Introduce abbreviations (MBT).

Response: MBT mice has been replaced with the expanded form i.e. MBT/Pas mice for clarity (II.298 and 300).

ll.293-306: How does that fit into the innate immune response chapter? Gets relevant for the subtopic from ll. 309 onward. The previous passage can be summarized in one sentence.

Response: The passage was modified accordingly and only the relevant statements have been included (II.310-312).

ll.335: In this sentence you indicate the humoral immune responses to specific RVFV proteins is studied because infection of animal models come along with high mortality and is therefore hard to study. Do consider that an infection of relevant animals (such as small ruminants) is barely lethal and I wouldnt agree that studies you describe following upon this sentence are performed based on this rational.

Response: The statements has been rephrased to bring in more clarity (II.340-343). Infection with wildtype RVFV strains routinely utilized in mouse models are often lethal across different strains. As a result, following adaptive response to such RVFV infections is challenging. Therefore, most of the knowledge related to adaptive immunity has come from studies which employed viral proteins or attenuated strains in mouse models, as stated in this review

References:

  1. Cartwright, H. N., Barbeau, D. J., & McElroy, A. K. (2020). Rift Valley Fever Virus Is Lethal in Different Inbred Mouse Strains Independent of Sex. Frontiers in microbiology11, 1962. https://doi.org/10.3389/fmicb.2020.01962
  2. Dodd KA, McElroy AK, Jones ME, Nichol ST, Spiropoulou CF. (2013). Rift Valley fever virus clearance and protection from neurologic disease are dependent on CD4+ T cell and virus-specific antibody responses. Journal of Virology. 87(11):6161-6171. DOI: 10.1128/jvi.00337-13.

II 354: Consider to have a look into this reference as well: doi: 10.3390/v14020350.

Response: relevant information from the suggested reference has been added to section 5, II.370-372.

Table 1: The vertical transmission listed for Clone 13 is also observed for MP12 vaccinations.

Response: The adverse effect of MP12 vaccination in early pregnancy was included in Table 1.

Consider to move ll.516-529 after l.499, this would enhance the reading flow.

Response: The paragraph was moved as per suggestion to II.515-528.

Reviewer 3 Report

Rift Valley Fever Virus virology, pathobiology, and vaccine candidates are reviewed in this manuscript.

Comments to authors:

1.       The current literature is well-represented except at the end of Section 3 regarding RVFV and autophagy. Several different groups have published studies regarding autophagy upregulation as a host response to RVFV infection.

2.       Figure 1 legend, line 234: Please correct the typographical error in “LRPI”. The correct designation is LRP1.

3.       Section 5, line 179: RIG-1 preferentially binds ssRNA (as stated in Figure 1 legend). Please clarify or correct the statement that RIG-I “…identifies 5’-triphosphate RNA and dsDNA…”. RVFV is a ssRNA virus and RIG-1 recognizes RVFV (line 188).

4.       Section 5, lines 293-294: Please add references for the studies in the 3 different species of mammals.

5.       Section 6 RVFV vaccine development: This section would benefit from subheadings to guide the reader through candidate vaccines, current MP12 vaccine, candidate vaccines, and viral vector systems.

6.       Section 6, lines 396-399: Add a reference to Caplen H, Peters CJ, Bishop DH. Mutagen-directed attenuation of Rift Valley fever virus as a method for vaccine development J. Gen. Virol. (1985) for the original development of the MP12 vaccine strain by the U.S Army Medical Research Institute for Infectious Diseases (USAMRIID).

7.       Section 6 lines 417-425: This section contains inaccurate information and should be checked and rewritten for clarity and accuracy. In the cited study, reference 123, only 6/10 calves had significant histopathological abnormalities. Lambs were not affected. Furthermore, viral antigen was detected in 1 calf. These key outcomes of reference 123 are misstated in this review manuscript.

8.       Abbreviations should be defined at first use for dpi and VN.

The quality of English is excellent. Please correct some minor grammar/typographical errors on lines 163, 175, 337, and 419.

Author Response

We thank the reviewer for the constructive feedback.

Comments to authors:

  1. The current literature is well-represented except at the end of Section 3 regarding RVFV and autophagy. Several different groups have published studies regarding autophagy upregulation as a host response to RVFV infection.

Response: Current literature on autophagy upregulation during RVFV infection is represented from lines 232-245. The sentences have been rephrased for more clarity.

  1. Figure 1 legend, line 234: Please correct the typographical error in “LRPI”. The correct designation is LRP1.

Response: We apologise for the error and the typography has been corrected to Lrp1 (revised document line 250).

  1. Section 5, line 179: RIG-1 preferentially binds ssRNA (as stated in Figure 1 legend). Please clarify or correct the statement that RIG-I “…identifies 5’-triphosphate RNA and dsDNA…”. RVFV is a ssRNA virus and RIG-1 recognizes RVFV (line 188).

Response: The statement in line figure 1 legend (lines 251-252) and in main text (lines 190-191) are modified for clarity. RIG-I recognizes short dsRNA formed as an intermediate of viral replication in RNA viruses, and uncapped, 5’-triphosphorylated ssRNA. This explains the recognition of ssRNA genome of RVFV by RIG-I.

  1. Section 5, lines 293-294: Please add references for the studies in the 3 different species of mammals.

Response: The impact of aerosolized RVFV particles in four different species of non-human primates if from a single study. The reference is added at the end of sentence in lines 310 and 312.

  1. Section 6 RVFV vaccine development: This section would benefit from subheadings to guide the reader through candidate vaccines, current MP12 vaccine, candidate vaccines, and viral vector systems.

Response: Additional subheadings have been added to the text  (lines 392-476)

  1. Section 6, lines 396-399: Add a reference to Caplen H, Peters CJ, Bishop DH. Mutagen-directed attenuation of Rift Valley fever virus as a method for vaccine development J. Gen. Virol. (1985) for the original development of the MP12 vaccine strain by the U.S Army Medical Research Institute for Infectious Diseases (USAMRIID).

Response: The reference to original paper by Caplen et. al. has been cited as suggested by the reviewer (line 409).

  1. Section 6 lines 417-425: This section contains inaccurate information and should be checked and rewritten for clarity and accuracy. In the cited study, reference 123, only 6/10 calves had significant histopathological abnormalities. Lambs were not affected. Furthermore, viral antigen was detected in 1 calf. These key outcomes of reference 123 are misstated in this review manuscript.

Response: We agree with the reviewer that the statements in the initial version is misleading and have now been rephrased with more accurate information in the revised manuscript(lines 426-430).

  1. Abbreviations should be defined at first use for dpi and VN.

Response: Abbreviations have been included at the first use for dpi (line 272) and VN (line 323). Additionally, expanded form and abbreviation for endoplasmic reticulum is provided at first use in line 128.